# RETRACTED: Long-Lasting Exendin-4-Loaded PLGA Nanoparticles Ameliorate Cerebral Ischemia/Reperfusion Damage in Diabetic Rats

**DOI:** 10.3390/jpm12030390

**Published:** 2022-03-03

**Authors:** Cheng-Hsun Chung, Shiu-Dong Chung, Yu-Hsuan Cheng, Chun-Pai Yang, Chiang-Ting Chien

**Affiliations:** 1Department of Life Science, School of Life Science, College of Science, National Taiwan Normal University, No. 88, Tingzhou Road, Taipei City 116, Taiwan; 60643024s@ntnu.edu.tw (C.-H.C.); 80943003s@ntnu.edu.tw (Y.-H.C.); 2Division of Urology, Department of Surgery, Far Eastern Memorial Hospital, New Taipei City 220, Taiwan; chungshiudong@gmail.com; 3Department of Nursing, College of Healthcare & Management, Asia Eastern University of Science and Technology, New Taipei City 220, Taiwan; 4General Education Center, Asia Eastern University of Science and Technology, New Taipei City 220, Taiwan; 5Department of Neurology, Kuang Tien General Hospital, No. 117, Shatian Road, Shalu District, Taichung City 433, Taiwan; 6Department of Nutrition, Huang-Kuang University, Taichung 433, Taiwan

**Keywords:** voiding function, diabetes, endoplasmic reticulum stress, stroke, apoptosis, autophagy, pyroptosis

## Abstract

Exendin-4 (Ex-4) is an incretin mimetic agent approved for diabetes treatment and neuronal protection. However, the required frequent injections restrict its clinical application. We prepared Ex-4-loaded poly(d,l-lactide-co-glycolide) nanoparticles (PEx-4) and investigated their effect on cerebral ischemia/reperfusion (IR) injury associated with micturition center damage-induced cystopathy in diabetic rats. Using ten minutes of bilateral carotid artery occlusion combined with hemorrhage-induced hypotension of the IR model in streptozotocin-induced type 1 diabetic (T1DM) Wistar rats, we compared the effects of Ex-4 and PEx-4 on prefrontal cortex edema, voiding function and oxidative stress including cerebral spinal fluid (CSF) reference H_2_O_2_ (RH_2_O_2_) and HOCl (RHOCl) levels, endoplasmic reticulum (ER) stress, apoptosis, autophagy and pyroptosis signaling in brain and bladder by Western blot and immunohistochemistry. Single injection of PEx-4 displayed higher CSF antioxidant activity and a long-lasting hypoglycemic effect compared to Ex-4 in rats. T1DM and IR primarily enhanced CSF RH_2_O_2_, and pIRE-1/caspase-12/pJNK/CHOP-mediated ER stress, caspase-3/PARP-mediated apoptosis, Beclin-1/LC3B-mediated autophagy and caspase-1/IL-1β-mediated pyroptosis signaling in the damaged brains. Our data further evidenced that PEx-4 were more efficient than Ex-4 in attenuating IR-evoked prefrontal cortex edema, the impairment in micturition center and the enhanced level of CSF RH_2_O_2_ and HOCl, ER stress, apoptosis, autophagy and pyroptosis parameters in the damaged brains, but had less of an effect on IR-induced voiding dysfunction in bladders of T1DM rats. In summary, PEx-4 with stronger antioxidant activity and long-lasting bioavailability may efficiently confer therapeutic efficacy to ameliorate IR-evoked brain damage through the inhibitory action on oxidative stress, ER stress, apoptosis, autophagy and pyroptosis signaling in diabetic rats.

## 1. Introduction

Diabetes mellitus (DM), resulting from defects in insulin secretion, insulin action, or both, is a metabolic disorder with multiple serious complications [1], possibly by hyperglycemia-induced oxidative stress and inflammation. It is known that the risk for stroke is doubled in DM patients in comparison with the general population, and these patients are at increased risk of death due to cerebrovascular diseases. The most common complications of DM are lower urinary tract symptoms (LUTS) including diabetic bladder dysfunction [2]. The occurrence of urinary incontinence is also a common sequela after acute hemispheric stroke [3]. Several urodynamic parameters from the ischemic and hemorrhagic stroke patients displayed an alteration in total bladder capacity, post-void residual urine volume and bladder compliance associated with detrusor overactivity and detrusor underactivity [4]. We suggested that increased oxidative stress by DM may susceptibly contribute to the induction of ischemic stroke and to impair the micturition center located in the brain, leading to a further impairment in voiding function.

In DM-induced global brain ischemia and related cystopathy, antidiabetic and neuroprotective drugs were considered [5]. GLP-1 (glucagon-like peptide-1) is a gut hormone and binds to a seven transmembraneous domain-G-protein-coupled receptor glucagon-like peptide 1 receptor (GLP-1R) to activate downstream cyclic AMP signaling [6]. GLP-1 has been suggested as a therapeutic drug in DM by decreasing blood glucose [7]. Exendin-4 (Ex-4 with amino acid sequence of HGEGTFTSDLSKQMEEEAVRLFIEWLK NGGPSSGAPPPS-NH2, 4186.7 Da), a peptide isolated from the salivary secretions of the lizard *Heloderma suspectum* (Gila monster), shares a 53% amino acid identity with GLP-1 [8,9]. Ex-4 has been shown to have several beneficial antidiabetic actions including glucose-dependent enhancement of insulin secretion, attenuation of gastric emptying, reduction in food intake, and increases in β-cell mass and β-cell function [10,11,12]. In addition, the hypoglycemic, cardiovasoprotective, and neuroprotective effects of Ex-4 have also been reported [13,14]. However, this drug is limited by a short half-life, high burst release, and lower bioactivity.

Recently, we have developed long-lasting Ex-4-loaded poly(D,L-lactide-co-glycolide) (PLGA) microspheres with a size around 15 ± 3.3 µm and found that the Ex-4 microsphere was more effective than traditional Ex-4 to improve bilateral carotid artery occlusion combined with hemorrhage-induced hypotension-induced oxidative injury and cognitive deficits through the activated p-Akt/p-eNOS and suppressed NF-κB/ICAM-1 signaling, ER stress, and apoptosis pathways [15]. PLGA is one of the most successfully used biodegradable nanosystems for the development of nanomedicines because it undergoes hydrolysis in the body to produce the biodegradable metabolite monomers, lactic acid and glycolic acid [16], and has been approved by the US FDA and European Medicine Agency in various drug delivery systems in humans. Since the body effectively deals with these two monomers, there is very minimal systemic toxicity associated with using PLGA for drug delivery or biomaterial applications [16]. Because we previously prepared Ex-4 microspheres with a large size of 15 ± 3.3 µm, a possible interruption of capillary blood flow may occur in brain microcirculation. In this study, we aimed to prepare PLGA-loaded Ex-4 nanoparticles (PEx-4) to determine whether PEx-4 is stronger than Ex-4 in bioavailability and long-lasting effect, which provide systemic protection against hypoxic/ischemic and hemorrhagic stroke-evoked brain and bladder injury. We also explored the effects of PEx-4 and Ex-4 on hypoxic/ischemic and hemorrhagic stroke-induced oxidative stress for the first time including cerebral spinal fluid H_2_O_2_ and HOCl amount and ER stress, apoptosis, autophagy and pyroptosis signaling in the impaired brain and bladder.

## 2. Materials and Methods

### 2.1. Ethical Approval

All surgical and experimental procedures were approved by the Institutional Animal Care and Use Committee of Far-East Memory Hospital (Approval number 106015 on the date of 6 June 2017) and were in accordance with the guidelines of the National Science Council of the Republic of China (NSC, 1997).

### 2.2. Chemicals

PLGA (cat. no. P2191, lactide/glycolide = 50/50, Mw = 30,000 to 60,000 Da), poly(vinyl alcohol) (87 to 89% hydrolyzed), polyvinyl alcohol (PVA), Ex-4, and dichloromethane were purchased from Sigma-Aldrich Chemical (St. Louis, MO, USA).

### 2.3. Experimental Design and Preparation of Ex-4-Loaded PLGA Nanoparticles

PLGA nanoparticles containing Ex-4 were prepared using a water-oil-water (*w*/*o*/*w*) emulsion solvent evaporation method as described previously [17]. Briefly, 1 mg of Ex-4 was dissolved in 200 μL distilled water. The aqueous solution was emulsified with 1 mL dichloromethane containing PLGA, using a MICCRA D-1 homogenizer (ART Prozess- & Labortechnik GmBH & Co KG, Müllheim, Germany) at 20,000 rpm for 180 s. The primary emulsion was then added to 5 mL of 3% polyvinyl alcohol (PVA) and sonicated for 60 s to form a double emulsion, using a Qsonica Sonicator (Q700 Ultrasonic Processor, Qsonica, LLC, Newtown, CT, USA). The resulting emulsion was combined with 50 mL of 0.5% PVA and stirred for 3 h at room temperature, allowing the dichloromethane to evaporate. The resulting PLGA nanoparticles were washed three times in distilled water, by centrifugation at 10,000× *g* (Centrifuge 5810R; Eppendorf AG, Hamburg, Germany).

### 2.4. Nanoparticles Characterization

We used a scanning electron microscope (JEOL-6500F, JEOL, Tokyo, Japan) to characterize the morphology and size of the PLGA nanoparticles. The images were analyzed using the ImageJ Windows 10 (32/64 bit) software (National Institutes of Health, NIH, Bethesda, Rockville, MD, USA).

### 2.5. In Vivo Drug Release Study

The in vivo drug release method was described in our previous report [15]. The Ex-4 and PEx-4 in saline were subcutaneously administered once to male Wistar rats at a dosage of 50 μg/kg. Blood was collected by retro-orbital bleeding into tubes containing EDTA at different time points for 7 days, transferred into 1.5 mL centrifuge tubes, and centrifuged at 15,000× *g* for 5 min. To measure Ex-4 entry into the brain parenchyma, we measured the Ex-4 concentration in the cerebrospinal fluid (CSF). Under anesthesia, 50 to 100 μL CSF was collected from the medullary cisterna magna of rats. The Ex-4 concentration in CSF and plasma was assayed with an Enzyme Immunoassay Kit (EK-070-94, Phoenix Pharmaceuticals, Burlingame, CA, USA).

### 2.6. Animals and Grouping

A total number of eighty female adult Wistar rats (200–250 g) with age > 8 weeks were purchased from BioLASCO Taiwan Co. Ltd. (I-Lan, Taipei, Taiwan) and housed at the Experimental Animal Center of Far-East Memory Hospital at a constant temperature (24 ± 1 °C) and with a consistent light cycle (light from 07:00 to 18:00 o’clock). All rats were maintained two per cage throughout the experiment. During the experimental periods, rats had free access to tap water and chow. An adaptation period of one week was allowed before the initiation of the experimental protocols. Food and water were provided ad libitum. Body weight was measured every week. On the last day of the experimental period, the animals were placed in metabolic cages to collect urine and some physiological parameters. All efforts were made to minimize both animal suffering and the number of animals used throughout the experiment. We used female rats due to the ease of recording of bladder function.

One previous study suggested that STZ-induced diabetic rats are the best model for the DM and induce diabetes similar to human [18]. Rats were made diabetic (DM) by intraperitoneal injection of streptozotocin (STZ) (60 mg/kg, Sigma-Aldrich, St. Louis, MO, USA). Rats with similar age and body weight were randomly assigned to one of the following 8 groups (*n* = 10 in each group): (1) control (PLGA vehicle-treated, Con) group, (2) DM (PLGA vehicle-treated, DM) group, (3) control + IR group (ConIR), (4) DM + IR group (DMIR), (5) control + IR + 50 μg/kg/week Ex-4 group (ConIR-Ex4), (6) control + 50 μg/kg/week Ex-4-loaded PLGA nanoparticles-treated group (ConIR-PEx4), (7) DM + IR + 50 μg/kg/week Ex-4-treatment group (DMIR-Ex4), and (8) DM + IR + 50 μg/kg/week PEx-4-treated group (DMIR-PEx4). The onset of DM occurred rapidly and was associated with polydipsia, polyuria, and a tail vein blood glucose concentration > 250 mg/dL (One Touch II; LIFESCAN, Milpitas, CA, USA). After two weeks of STZ, Ex-4 or PEx-4 was administered once per week via subcutaneous injection (50 μg/kg/week) for another two weeks. In this study, we excluded the rats that died during diabetes induction or global cerebral ischemia (IR) injury.

### 2.7. Global Cerebral Ischemia

In groups 3–8, temporary global cerebral ischemia (IR) was induced under avertin (2,2,2-tribromoethanol) anesthesia by 10 min bilateral common carotid arteries with hemorrhage-induced hypotension according to our previous report [15]. In brief, animals were anesthetized by avertin (0.02 mL/g) during surgery to minimize discomfort and were fixed on an operating table. The left and right femoral arteries were catheterized with a PE-50 catheter, respectively, to continuously record arterial blood pressure and blood sampling. After heparinization, blood was quickly withdrawn via the femoral artery. When the mean arterial blood pressure reached 30 mmHg, the bilateral common carotid arteries were clamped with surgical clips for 10 min, after which the clips were removed and blood was reinfused via the femoral vein. In the sham-operated animals, the vessels were exposed, but neither blood withdrawal nor clamping of the carotid arteries was performed. The rectal temperature was maintained at 37 ± 0.5 °C in all animals during surgery with a homeothermic blanket. After the experiment, the animals were sacrificed by intravenous KCl.

### 2.8. Cerebral Edema Measurement by T2-Weighted Magnetic Resonance Imaging (MRI)

MRI was carried out in the animals using a Bruker Biospec 7-T MRI system as described previously [15]. Anesthesia was induced with 5% halothane and maintained with 1.5% halothane (both concentrations prepared in O_2_:N_2_O, 70:30 by volume). Rats were intubated and mechanically ventilated at a rate of 60 breaths/min. Heart rates and respiratory rates were monitored throughout the procedure, and body temperature was maintained at 37 °C. A rapid-acquisition relaxation enhancement T2-weighted sequence was used to determine the precise lesion location, with a rapid-acquisition relaxation enhancement factor (RARE) of 16, a repetition time of 5086 ms, and an echo time of 70.1 ms. The in-plane resolution was 250 × 250 × 250 µm and 15 slices were acquired. A second T2-weighted image set of 25 contiguous slices was acquired at the lesion site (RARE factor = 16, repetition time = 5086 ms, echo time = 70.1 ms) with an in-plane resolution of 117 × 117 × 500 µm. Infarct areas were manually delineated on the MRI images using Paravision software (Bruker Corp., Billerica, MA, USA) and multiplied by the interslice distance to calculate the infarct volume [15]. Image J was used to analyze the area of brain edema as described in Figure 1B.

### 2.9. Measurement of Specific CSF ROS Activity

Hydrogen peroxide (H_2_O_2_) and hypochlorite (HOCl) are two major ROS generated from activated neutrophils via the myeloperoxidase (MPO) [19]. In this part of the study, 100–200 μL of cerebrospinal fluid (CSF) was withdrawn from the cisterna magna in the urethane-anesthetized (1.2 g/kg, Sigma, Missouri, KC, USA, intraperitoneally) rats. We measured CSF H_2_O_2_ and HOCl amount by an amplified chemiluminescence (CL) technique, as described previously [19]. In brief, CL signals emitted from the “test mixture” of cerebral spinal fluid (CSF) (or phosphate-buffered saline (PBS) (50 mmol/L, pH 7.4) as a background control), H_2_O_2_ (or HOCl), and CL-emitting substance (i.e., luminol (5-amino-2,3-dihydro-1,4-phthalazinedione); Sigma, Chemical Co., St. Louis, MO, USA) were measured with a multi-wavelength CL spectrum analyzer (CLA-SP2, Tohoku Electronic Ind., Co., Sendai, Japan). In this study, we used 25 μL of CSF sample or 25 μL of PBS throughout. We first mixed 25 μL of CSF and 1.0 mL of 25 μmol/L luminol in a 4.0 mL quartz cell (1 × 1 × 4 cm) for 100 s. Next, 1.0 mL of 0.03% H_2_O_2_ or 0.012% NaOCl was immediately added into the quartz cells. Luminol stock solution (250 μmol/L) was prepared as 1 mg of luminol dissolved in 22.7 mL of PBS. The CL emitted from the above reaction mixture was recorded and measured as “reference H_2_O_2_ counts” (RH_2_O_2_) or “reference HOCl counts” (RHOCl). PBS was added to the test system, and the RH_2_O_2_ and RHOCl yielded were recorded as the background counts. A RH_2_O_2_ or RHOCl level indicated the ROS value.

### 2.10. Cystometric Parameters

Under anesthesia, PE-50 catheters were placed in the left femoral artery for measurement of ABP and in the left femoral vein for administration of drugs. ABP was recorded in an ADI system (Power-Lab/16S, ADI Instruments, Pty., Ltd., Castle Hill, Australia) with a transducer (Gould-Statham, Quincy, IL, USA). Body temperature was kept at 36.5–37 °C by an infrared light and was monitored with a rectal thermometer.

We introduced a transcystometric model to evaluate the micturition alteration in the bladder in response to IR injury. The method has been well-established as described previously in our lab [20]. Briefly, these rats were anesthetized by subcutaneous injection of urethane (1.2 g/kg body weight). After the bladder was exposed through a midline incision of the abdomen, a PE-50 catheter (bladder catheter) was inserted through the apex of the bladder dome and was connected via a T-tube to a P23 ID infusion pump and pressure transducer (Gould-Statham, Quincy, IL, USA). The intravesical pressure was recorded continuously in an ADI system (Power-Lab/16S, ADI Instruments, Pty., Ltd., Castle Hill, Australia). The following parameters of bladder responsiveness were measured: intercontraction interval (ICI), the time lag between two micturition cycles identified with active contractions (>10 mmHg), baseline bladder pressure (BP), micturition duration (MD), and contractile amplitude (Am = maximal bladder pressure—BP) for a micturition. The cystometric parameters were evaluated during each 8 min interval.

### 2.11. Immunohistochemistry

The method for immunohistochemistry was described previously [15]. In brief, tissue sections were deparaffinized in xylene and rehydrated in an ethanol series. The tissue sections were subjected to an antigen retrieval step. The buffer solution used for heat-induced epitope retrieval was sodium citrate buffer (10 mM sodium citrate, 0.05% Tween 20, pH 6.0). After 15 min of antigen retrieval step, the sections were blocked for non-specific binding with 5% bovine serum albumin (Sigma-Aldrich, St. Louis, MO, USA) for 1 h at room temperature and incubated with the primary antibodies for 18 h at 4 °C. The tissue sections were washed with PBS three times and then were incubated with secondary antibodies Alexa Fluor488 (1:200; Abcam, Cambridge, UK) and nuclear staining dye Hoechst33342 (1:1000; Sigma-Aldrich, St. Louis, MO, USA) for 1 h at room temperature. After washing with PBS, the tissue sections were mounted in mounting medium (Leica, Wetzlar, Germany). The slides were scanned by a Leica TCS SP3 laser confocal microscope (Leica) to obtain the confocal images. Primary antibodies including mouse anti-CHOP (1:100; Cell Signaling Technology, Denver, MA, USA), rabbit anti-Caspase-3 (1:100; Abcam), rabbit anti-Caspase-1 (1:100; Abcam) and rabbit anti-LC3β (1:100; Abcam) were used in this study. Bladder sections were also stained with hematoxylin and eosin or Masson’s trichrome for pathophysiologic evaluation.

### 2.12. Western Blotting

The tissues were ground to powder with a prechilled mortar and pestle. The tissue powder was lysed in Lysis Buffer (Cell Signaling Technology, Denver, MA, USA) supplemented with protease inhibitor for 10 min at 4 °C. The tissue homogenate was centrifuged at 14,000 rpm for 30 min. After centrifugation, the supernatant was collected into a new Eppendorf. The concentrations of proteins were measured by Protein Assay Dye Reagent Concentrate (Bio-Rad, Hercules, CA, USA). Then, the 40 μg protein samples were mixed with 1 × sample buffer and were boiled for 5 min. The protein samples were resolved in 10% SDS-polyacrylamide gel electrophoresis (SDS-PAGE) and transferred to PVDF membrane (Millipore, Billerica, MA, USA). Then, the blot was blocked with Hyblock (Hycell, Taipei, Taiwan) for 1 min, and incubated with primary antibody overnight at 4 °C. After washing three times with TBS, the blot was incubated with horseradish peroxidase (HRP)-conjugated secondary antibodies at room temperature for 1 h.

Detection of signals was performed by Immobilon Western Chemiluminescent HRP Substrate (Millipore). Primary antibodies, including pIRE-1, IRE-1, Caspase-12, pJNK, JNK, CHOP, IL1β, Caspase-1, LC3β, Caspase-3, PARP, and β-actin, were used. Secondary antibodies including HRP-conjugated rabbit anti-mouse IgG, HRP-conjugated donkey anti-goat IgG, and HRP-conjugated goat anti-rabbit IgG (all for 1:10,000; all from Sigma-Aldrich, St. Louis, MO, USA) were used in this study.

### 2.13. Histologic Staining

A portion of the brain or bladder tissue was cut and fixed in 10% neutral buffered formalin solution, dehydrated in graded ethanol, and embedded in paraffin. Sections (4 μm) of the bladder or brain were stained with hematoxylin and eosin to evaluate the extent of morphological changes. Then, 4 µm sections of formalin-fixed bladders were stained with Masson’s trichrome for fibrosis staining (blue collagen accumulation).

### 2.14. Statistical Analysis

All values and data were expressed as mean ± standard error mean (SEM). Data recording was not blinded because of technical requirements, but data analyses were blinded for biochemical analyses. Statistical analyses were performed using one-way ANOVA, followed by Bonferroni’s post hoc test. Differences between groups were considered statistically significant when *p* < 0.05. Statistical analyses were performed using the SPSS 18.0 software (IBM, Armonk, NY, USA; IBM SPSS Modeler Knowledge Center) and curve fitting was carried out using GraphPad Prism (v. 6.0, GraphPad Software, La Jolla, CA, USA).

## 3. Results

### 3.1. PEx-4 Exerts a More Long-Lasting Hypoglycemic Effect than Ex-4

Our prepared PEx-4 were approximately 68.0 ± 3.2 nm in size (Figure 1A). After subcutaneous injection in vivo, PEx-4- and Ex-4-treated rats showed a peak plasma level at 2 h (Figure 1B). The peak plasma Ex-4 concentration after 2 h of Ex-4 treatment was significantly higher (*p* < 0.05) than that of PEx-4 treatment, implicating a large burst release by Ex-4. From 6 h to 7 days, the plasma Ex-4 concentration of the PEx-4 group was maintained at a stable level and was higher (*p* < 0.05) than that of the Ex-4 group, indicating sustained release of PEx-4 (Figure 1C). Our data showed that the CSF Ex-4 level was consistent with the plasma Ex-4 value (Figure 1D). Two hours after Ex-4 treatment, the peak CSF Ex-4 value was higher (*p* < 0.05) than that of PEx-4 treatment, also implicating a great surge of Ex-4 release. The CSF Ex-4 value in the PEx-4 group was well maintained and higher (*p* < 0.05) than that of the Ex-4 group 6 to 24 h after treatment, indicating sustained release of Ex-4 into the plasma and brain from PEx-4. Subcutaneous administration of Ex-4 or PEx-4 significantly decreased blood glucose 6 h after treatment, but PLGA vehicle nanoparticles did not produce hypoglycemic effect (Figure 1E). PEx-4 significantly decreased blood glucose levels from 12 h to 7 days after treatment in DM rats, but Ex-4 had no hypoglycemic effect on DM-induced hyperglycemia from 12 h to 7 days after treatment. These data demonstrated that PEx-4 exerts a more long-lasting hypoglycemic effect than Ex-4.

### 3.2. PEx-4 Is More Efficient than Ex-4 on Reduction IR-Enhanced RH_2_O_2_ and RHOCl Activity in CSF

In Figure 2, the typical data of RH_2_O_2_ and RHOCl response from eight groups of rats 24 h after IR injury are displayed. We noted that DM elevated a higher value of RH_2_O_2_, not RHOCl, in the basal state. CSF RH_2_O_2_ and RHOCl in ConIR and DMIR were significantly increased vs. Con and DM groups. However, the enhanced response of RH_2_O_2_ and RHOCl was partly decreased by Ex-4 or PEx-4 treatment. PEx-4 was more efficient than Ex-4 in reducing DM- and IR-induced oxidative stress.

### 3.3. PEx-4 Is More Efficient than Ex-4 on Decreasing IR-Induced Brain Edema

Previous studies had displayed the neuroprotective effect of Ex-4 on reducing brain injury in cerebral ischemic rats [14,15] and peripheral neuropathy in STZ-induced diabetes [21]. We compared the neuroprotective effect of PEx-4 and Ex-4 in DM and IR injury. The brain edema formation after IR was examined as demonstrated in Figure 3A, reflected by T2-weighted images and analyzed by the thresholding tool of Image J freeware in Figure 3B. On T2-weighted images, the bright zone in the brain cortex was regarded as the edema region. Through MRI data, the edema region in the CT1 (Figure 3C), CT2 (Figure 3D) and CT3 (Figure 3E) sections was significantly increased in ConIR and DMIR rats. The increased degree of brain edema by IR was significantly depressed with Ex-4 or PEx-4 in IR-Ex4 and IR-PEx4 rats (Figure 3C–E). PEx-4 in the IR-PEx4 group was more efficient than Ex-4 in the IR-Ex4 group at decreasing IR-induced CT1 edema (Figure 3C). We also demonstrated the paralleled histologic evidence of the CT1 section of these eight groups of brains in Figure 3F. In the edema area, the shrunken neuron and vacuoles were found especially in DM brains with IR injury.

### 3.4. PEx-4 Was More Efficient than Ex-4 on Reducing IR-Enhanced Neuronal Shrinkage and ER Stress in Brains

The appearance of neuronal shrinkage and vacuolization in rat brains after IR has been reported previously [22]. In the prefrontal cortex of Con and DM brains, most neurons appeared to be normal (size > 12 µm) with well-defined nuclei and clearly visible cytoplasm (Figure 4A). After IR, the morphological abnormalities, including shrinkage of neurons (size < 10 µm), vacuolization and angular with dark-stained nuclei, were observed in ConIR and DMIR brain (Figure 4A). The quantitative data were acquired by counting the number of shrunken neurons in the damaged areas, which were chosen randomly on the slides. The number of shrunken neurons was significantly (*p* < 0.05) decreased with the Ex-4 or PEx-4 treatment in IR-Ex4 and IR-PEx4 brains (Figure 4B). PEx-4 was more efficient than Ex-4 in decreasing the number of shrunken neurons.

It had been proved that ER stress plays an important role in mediating ischemic neuronal cell death [23]. ER stress-associated protein CHOP was found in Con and DM brains (Figure 4C). IR significantly (*p* < 0.05) increased the CHOP-positive stained cells in ConIR and DMIR groups as compared to respective Con and DM. As compared to IR groups, Ex-4 and PEx-4 treatment significantly (*p* < 0.05) decreased the enhanced CHOP-positive cells in the IR-Ex4 and IR-PEx4 rats (Figure 4D).

### 3.5. PEx-4 Was More Efficient than Ex-4 on Decreasing ER Stress-, Apoptosis-, Pyroptosis- and Autophagy-Related Protein Expression in Rat Brain with DM or IR Injury

The protein profile of the ER stress-, apoptosis-, pyroptosis- and autophagy-associated proteins in rat brain was analyzed by Western blotting (Figure 5A). ER stress-associated proteins including pIRE-1/IRE-1 (Figure 5B), Caspase-12 (Figure 5C), pJNK/JNK (Figure 5D), CHOP (Figure 5E), Beclin-1 (Figure 5F), LC3B (Figure 5G), pyroptosis-related Caspase-1 (Figure 5H), IL-1β (Figure 5I) and apoptosis-related Caspase-3 (Figure 5J), except PARP (Figure 5K), were significantly enhanced in DM brains vs. Con brains. IR significantly enhanced pIRE-1/IRE-1, Caspase-12, pJNK/JNK, CHOP, Caspase-1, Beclin-1, LC3B, cCaspase-3 and PARP in the ConIR vs. Con brains. IR significantly increased LC3B, IL-1β and PARP in the DMIR group vs. DM group. As compared to ConIR, Ex-4 treatment significantly (*p* < 0.05) depressed CHOP, IL-1β, Caspase-3 and PARP in IR-Ex4 groups. In comparison with the DMIR group, Ex-4 treatment significantly (*p* < 0.05) decreased CHOP, IL-1β, and Caspase-1 in the DMIR-Ex4 groups. PEx-4 treatment significantly inhibited CHOP, IL-1β, Caspase-1 and Caspase-3 in the IR-Ex4 group vs. IR group. Our data further demonstrated that PEx-4 in IR-PEx4 group was more efficient than Ex-4 in the IR-Ex4 group in inhibiting pIRE-1/IRE-1, Caspase-12, pJNK/JNK, CHOP, Caspase-1, IL-1β, Beclin-1, LC3B, Caspase-3 and PARP in the IR brains.

### 3.6. PEx-4 Was More Efficient than Ex-4 in Depressing Pyroptosis, Autophagy and Apoptosis Immunofluorescent Staining in DM or IR Brains

With the immunofluorescent staining technique, we explored the effect of DM, IR and Ex-4 treatment on the brain expression of pyroptosis-related Caspase-1, autophagy-related LC3B and apoptosis-related Caspase-3 expression in these eight groups of rat brains. The green fluorescent density of these three markers was demonstrated in Figure 6A. The baseline level of Caspase-1, LC3B and Caspase-3 expression was less expressed in the Con brains. DM significantly (*p* < 0.05) enhanced brain Caspase-1 (Figure 6B), LC3B (Figure 6C) and Caspase-3 (Figure 6D) stains vs. the Con group, whereas IR further (*p* < 0.05) enhanced Caspase-1, LC3B and Caspase-3 stain in the ConIR and DMIR brains vs. respective Con and DM brains. Ex-4 and PEx-4 significantly (*p* < 0.05) reduced DM- and IR-enhanced Caspase-1, LC3B and Caspase-3 fluorescence in the brains of IR-Ex4 or IR-PEx4 groups. PEx-4 was more efficient than Ex-4 in reducing IR-induced Caspase-3-mediated apoptosis, LC3B-mediated autophagy and Caspase-1-mediated pyroptosis in the ConIR brains through its long-lasting neuroprotection effect.

### 3.7. Ex-4 and PEx-4 Produced Less Protective Effect on Cystometry in Eight Groups of Rats

The representative cystometric graphs and the measurement of four urodynamic parameters for the eight groups of rats are shown in Figure 7A,B. The cystometry statistical data are indicated in Figure 7C–F. The micturition interval ICI (Figure 7C), MD (Figure 7E) and BP (Figure 7F) were significantly (*p*  <  0.05) increased in the DM group as compared to the Con group. The level of Am in the DM group was not affected by four-week DM induction as compared to the Con group (Figure 7). IR injury depressed voiding function and caused urine retention in the ConIR and DMIR bladders associated with the decreased ICI and Am and increased MD and BP. The treatment of PEx-4 in IRPE and Ex-4 in the IRE group displayed a similar urodynamic response for ICI (Figure 7C), Am (Figure 7D), MD (Figure 7E) and BP (Figure 7F) vs. respective IR groups implicating less protection conferred by PEx-4 or Ex-4 on bladder dysfunction.

### 3.8. IR, Ex-4 or PEx-4 Had No Significant Effect on Bladder Fibrosis in Eight Groups of Rats

Our data found that the size and weight of DM bladders were larger than those of Con bladders. Hematoxylin and eosin staining revealed that the lamina propria layers in DM bladders were thicker than those in Con bladders (Figure 8A). The thickening region in DM bladder was determined by Masson’s trichrome method. Blue-stained collagen fibers (fibrosis) were significantly increased in DM bladder (Figure 8B), possibly contributing to DM bladder hypertrophy. By use of Image J to analyze the degree of bladder fibrosis (Figure 8C), IR, Ex-4 or PEx-4 had no significant effect on blue collagen accumulation (fibrosis) (Figure 8D).

### 3.9. PEx-4 Did Not Produce Better Protection than Ex-4 on Stress-Associated Proteins in Rat Bladders with DM or IR Injury

The protein profile of ER stress-, apoptosis-, pyroptosis- and autophagy-associated proteins in the eight groups of rat bladders was analyzed by Western blotting (Figure 9A). ER stress-associated proteins including pIRE-1/IRE-1 (Figure 9B), cleaved Caspase-12 (Figure 9C), pJNK/JNK (Figure 9D), CHOP (Figure 9E), pyroptosis-related Caspase-1 (Figure 9F), IL-1β (Figure 9G), Beclin-1 (Figure 9H), LC3B (Figure 9I), and apoptosis-related cCaspase-3 (Figure 9J) and PARP (Figure 9K) were significantly (*p* < 0.05) up-regulated in DM as compared to Con. IR in the ConIR group significantly (*p* < 0.05) activated pIRE-1, cleaved Caspase-12, pJNK, CHOP, IL-1β, Caspase-1, Beclin-1, LC3B, Caspase-3 and PARP vs. Con group. DMIR significantly enhanced CHOP, Caspase-1, and Caspase-3 expression vs. DM. As compared to ConIR or DMIR group, Ex-4 significantly (*p* < 0.05) down-regulated CHOP, IL-1β, Caspase-1 and PARP expression in the ConIR and DMIR groups. PEx-4 significantly (*p* < 0.05) down-regulated pJNK, CHOP, IL-1β, Caspase-1 and LC3B expression in the ConIR-PEx4 and DMIR-PEx4 groups vs. respective ConIR and DMIR groups. These data also informed that PEx-4 was not significantly more efficient than Ex-4 in reducing DM- or IR-enhanced ER stress, pyroptosis, autophagy and apoptosis in the bladders.

### 3.10. PEx-4 Was More Efficient than Ex-4 in Reduction of Apoptosis, Not Pyroptosis and Autophagy in Bladders with DM and IR Injury

We also used the immunofluorescent staining technique to determine the effect of DM, IR, Ex-4 and PEx-4 treatment on the bladder expression of pyroptosis-related Caspase-1, autophagy-related LC3B and apoptosis-related Caspase-3 expression in these eight groups of rat bladders. The green fluorescent density of these three markers is demonstrated in Figure 10A. The baseline level of Caspase-1, LC3B and Caspase-3 expression was expressed less in the Con bladders. DM significantly (*p* < 0.05) enhanced bladder Caspase-1 (Figure 10B), LC3B (Figure 10C) and Caspase-3 (Figure 10D) stains vs. the Con group, whereas IR further (*p* < 0.05) enhanced Caspase-1, LC3B and Caspase-3 stain in the ConIR and DMIR bladders vs. respective Con and DM bladders. Ex-4 and PEx-4 treatment efficiently and significantly (*p* < 0.05) reduced ConIR- or DMIR-enhanced Caspase-1, LC3B and Caspase-3 fluorescence in the bladders. These data further implicated that PEx-4 is more efficient than Ex-4 in reducing DMIR-induced Caspase-3-mediated apoptosis, not LC3B-mediated autophagy and Caspase-1-mediated pyroptosis in the bladders of IR-Ex4 and IR-PEx4 groups. According to our data, there was no significant difference in the bladder smooth muscle layer among these eight groups, implicating DM, IR, Ex-4 or PEx4 as having no direct effect on the bladder smooth muscle tissue.

## 4. Discussion

The increasing use of nanomaterials in a variety of industrial, commercial, and medical products and their environmental spreading have raised concerns regarding their potential toxicity to human health. For example, titanium dioxide nanoparticles may induce ultrastructural alterations appearing at the third week by inducing activation of caspase-3 and increasing intracellular ROS and DNA damage [24]. On the other hand, PLGA nanoparticles were beneficial for ensuring structural stability, enhancing absorption, followed by sustained drug release [25]. Biodegradable nanoparticles such as carbohydrate polymeric PLGA are frequently used to improve the therapeutic value of various water-soluble/insoluble medicinal drugs and bioactive molecules by improving bioavailability, solubility and retention time [15]. A specific drug loaded with PLGA nanoparticles using the single emulsion–solvent evaporation method possessed high drug loading content, encapsulation efficiency, and a small size, and negative surface charge contributed to a specific intracellular accumulation [26]. Previous results also suggest that PLGA nanoparticles represent a high potential for utilization in anticancer therapy due to their effectiveness and safety [27]. PLGA nanoparticles loaded with miR-30b-5p can improve cardiac function, attenuate myocardial injury, and regulate the expression of factors associated with cardiac hypertrophy and inflammation by targeting TGFBR2 [28]. PLGA nanoparticles are well known as promising delivery systems, especially in the area of nose-to-brain delivery by transcellular, intracellular and paracellular transport [29]. Intracellular uptake was observed for 80 and 175 nm within only 5 min after application to the epithelium. This information implicates that PLGA nanoparticles with high efficacy, small size, no toxicity and easy uptake by cells are excellent delivery systems.

Our data evidenced that PEx-4 using PLGA as a carrier conferred a long-lasting effect on hypoglycemia, antioxidant activity and neuroprotection against DM- and IR-induced ER stress and three types of programmed cell death. Our data informed that the peak value of Ex-4 is around 2 h and the half life is around 4 h, whereas the peak value of PEx-4 is around 2–3 h and the half life is almost 24–168 days by the plasma Ex-4 assay. Abnormalities in water balance have been regarded as an important role in the pathophysiology of traumatic brain injury or stroke [30]. Cerebral edema, defined as an abnormal increase in brain water content, resulted in an increase in intracranial pressure, ischemia and death. Through the MRI data, the edema that resulted from IR injury was restricted in the rat cortex region. Additionally, the severity of edema was increased gradually in a caudo-rostral pattern (Figure 1). Our data, presented in Figure 4, demonstrated that the shrunken neurons and CHOP-positive cells appeared in the brain edema or injury of prefrontal cortex. Our histologic data showed that histologic changes (shrunken neurons and vacuoles) in these areas increased brain ROS, CSF ROS amount, CHOP positive cells and NADPH oxidase gp91 expression [15], implicating a possibly oxidative stress-mediated cytotoxic edema in the MRI image. In addition, the shrunken neurons could be the DNA condensation of apoptotic bodies; however, this needs further experiments such as TUNEL stain to confirm the hypothesis. It had been reported that Ex-4 was able to enhance NGF-induced neuronal differentiation and attenuate neural degeneration following NGF withdrawal [31]. The neuroprotection of Ex-4 was also regarded as a potential therapeutic target in neurodegenerative disease [32] and our data further confirmed PEx-4 was more efficient in reducing ischemic brain-induced edema and injury, which may be related to aquaporin-4-mediated water uptake and compression of the adjacent capillary lumen [33]. Since the general behavioral outcomes after brain IR injury with Ex-4 microspheres treatment have been reported in our previous report [15], we examined another physiological parameter voiding response in these rats with brain IR injury treated with Ex-4 or PEx-4 nanoparticles in this study. The present study used carbohydrate polymer PLGA nanoparticles delivered Ex-4 to the animal in vivo and discovered the therapeutic effect on IR-induced brain injury and partial voiding dysfunction in DM rats. Pharmacokinetic evidence showed that PEx-4 was better than Ex-4 in terms of long-lasting release of Ex-4 in plasma and CSF, reduction in DM-evoked hyperglycemia, and decrease in oxidative stress, ER stress, apoptosis, autophagy and pyroptosis in the IR brain. In addition, our unpublished data indicated the dose of PEx-4 or Ex-4 did not have nephrotoxicity to impair renal tissue or renal function because the serum creatinine was not increased. Since Ex-4 was administered weekly despite the very short half life, the maximum efficacy of this drug might not be fully achieved in this study. A dosage of 50 μg/kg/week used in this study could be low and could not obtain adequate protective efficacy in this study. Thus, the superior efficacy of PEx-4 compared to Ex-4 with different dosages or treatment frequency requires further studies to conclude their therapeutic potential.

It has been revealed that lesions of the medial prefrontal cortex caused nocturnal incontinence, urinary urgency and frequent micturition and only rarely caused urinary retention [34,35]. In our study, the ischemic brain with the edema of prefrontal cortex disturbed micturition and maintenance of urinary continence in rat. Furthermore, lesioning the medial prefrontal cortex would prolong the time interval between volume-evoked bladder contractions without changing the amplitude of contractions [36]. Our brain IR injury-induced bladder dysfunction data were consistent with these previous studies. We evaluated the ROS amount in the IR brain and found a significant increase in ROS amount in the CSF, indicating the toxic effect of ROS in the damaged brain. The two major ROS generated from activated neutrophils via the myeloperoxidase system are hydrogen peroxide (H_2_O_2_) and hypochlorite (HOCl) [19], which are important mediators in oxidative stress and inflammation. Our data found that DM and IR injury increased H_2_O_2_ and HOCl in the CSF, implicating neuronal inflammation and oxidative stress. The novel finding indicated that increased ROS in CSF may impair brain function and the micturition center, leading to voiding dysfunction. A crosstalk role of exacerbated ROS production may occur between the brain and bladder. Increased excess oxidative stress that evoked neural and vascular injury contributed to DM-evoked stroke associated with voiding dysfunction [2,3,4,5]. Exacerbated ROS production leads to tissue damage through ROS-evoked abnormal signal transduction, inflammatory monocyte/macrophage infiltration, cellular dysfunction and programmed cell death (autophagy, apoptosis and pyroptosis) in several kinds of cells. ROS generation from the NADPH (nicotinamide adenine dinucleotide phosphate) oxidase or mitochondria may induce apoptosis, autophagy, or pyroptosis, via execution by caspases, lysosomal proteases, or endonucleases [37]. The increased ROS trigger apoptosis by activating Bax expression/caspase 3 activity/poly-(ADP-ribose)-polymerase (PARP) fragments, enhance autophagy by activating Beclin-1/Atg5-Atg12/LC3-II pathway and/or evoke pyroptosis via the activation of caspase 1/IL-1β signaling [37]. In our Western blot and immunofluorescent data, caspase-1 mediated pyroptosis, LC3B-mediated autophagy and caspase-3-mediated apoptosis were significantly enhanced in the IR brain and DM bladder. Our prepared PEx-4 was more efficient than Ex-4 in reducing these oxidative stresses (CSF ROS), ER stress and three types of programmed cell death in IR- and DM-induced injury, implicating the therapeutic potential of PEx-4 in the reduction in oxidative stress, ER stress, apoptosis, autophagy and pyroptosis.

ER stress was the essential step in progression of ischemic stroke. Oxygen and glucose deprivation from IR injury contributed to ER stress [38]. After IR injury, the abnormalities of the brain section including shrinkage of neurons and vacuolization were observed in our study. In addition, CHOP, as a downstream molecule in the ER-stress pathway [39], was highly expressed in IR and DM injury. Our data from brains found that IR-enhanced ER stress including enhanced expression in pIRE-1/IRE-1, Caspase-12, pJNK/JNK, and CHOP was reduced by the treatment of Ex-4 and PEx-4. In one previous study, Ex-4 was proved to attenuate ER stress through silent mating type information regulation 2 homolog1, which decreased by hyperglycemia [40,41]. Hyperglycemia also downregulated the expression of GLP-1R [42]. Since Ex-4 binds to GLP-1R and exerts biological functions, the downregulation of GLP-1R by hyperglycemia would attenuate the protection of Ex-4. A continuous release and long-lasting effect of Ex-4 by PEx-4 may trigger Ex-4/GLP-1R signaling to decrease ER stress. It is well known that Ex-4 protected the organs and cells from oxidative damage induced by DM [43,44,45,46]. These Ex-4 molecules are small enough to cross the blood–brain barrier [14,47], and GLP-1Rs are widely expressed throughout the brain. Our previous report delineated that Ex-4 in PLGA microspheres treatment reduces stroke-induced frontal cortex edema, ER stress, apoptosis, and upregulation of aquaporin 4, glial fibrillary acid protein, and ICAM-1 in the damaged brain [15], implicating its neuroprotection. Our present data further evidenced the neuroprotective effect of PEx-4 nanoparticles on the reduction in autophagy and pyroptosis in the DM brain.

The stroke patients with dysregulated bladder function underlying DM were taking anticholinergic medications to improve voiding dysfunction [48]. Three major mechanisms are possibly responsible for post-stroke-evoked urinary incontinence: (1) disruption of the neuromicturition pathways, resulting in bladder hyper-reflexia and urgency incontinence; (2) incontinence due to stroke-related cognitive and language deficits, with normal bladder function; and (3) concurrent neuropathy or medication use, resulting in bladder hyporeflexia and overflow incontinence [48]. Our present data confirmed that DM or cerebral IR injury led to voiding dysfunction in these rats. In this study, we further investigated the molecular mechanism and pathophysiological functions in DM bladder after cerebral IR damage and evaluated the therapeutic potential of PEx-4. Previous reports evidenced that Ex-4 could rescue bladder dysfunction that resulted from DM and IR injury [49,50]. Our data further discovered that several stress markers (ER stress, autophagy, apoptosis and pyroptosis) were partly downregulated in DM bladder with the treatment of PEx-4. However, we did not find a significant effect to ameliorate morphological alteration in DM bladders with the treatment of PEx-4. We suspect that long-term STZ induction might be the critical cause of severe and irreversible damage in the DM bladder. The suppressed DM- and IR-induced oxidative stress in ER stress, apoptosis, autophagy and pyroptosis was primarily found in the brain, not in the bladder, suggesting PEx-4’s target protection in brains not in bladders. On the other hand, PEx-4 displayed a smaller protective effect on bladder dysfunction, possibly due to a single treatment or a lower dosage of PEx-4. An increasing treatment frequency or dosage of PEx4 may confer efficient protection; however, it requires further experiments to be determined.

## 5. Conclusions

According to our data, T1DM and IR significantly enhanced CSF RH_2_O_2_, and pIRE-1/caspase-12/pJNK/CHOP-mediated ER stress, caspase-3/PARP-mediated apoptosis, Beclin-1/LC3B-mediated autophagy and caspase-1/IL-1β-mediated pyroptosis signaling in the damaged brains. PEx-4 were more efficient than Ex-4 in attenuating IR-evoked prefrontal cortex edema, the impairment in the micturition center and the enhanced level of CSF RH_2_O_2_ and HOCl, ER stress, apoptosis, autophagy and pyroptosis parameters in the damaged brains, but has less of an effect on IR-induced voiding dysfunction in bladders of T1DM rats. In conclusion, these results suggest that PEx-4 with stronger antioxidant activity and long-lasting bioavailability may efficiently confer therapeutic efficacy to ameliorate IR-evoked brain damage through the inhibitory action on oxidative stress, ER stress, apoptosis, autophagy and pyroptosis signaling in diabetic rats. We implicate that long-lasting release of GLP-1 agonists PEx-4 may confer therapeutic potential to treat cerebral IR injury in the DM subjects in clinical trials.

## Figures and Tables

**Figure 1 jpm-12-00390-f001:** (**A**) Morphology and size of exendin-4 (Ex-4)-loaded poly(D,L-lactide-co-glycolide) (PLGA) nanoparticles (PEx-4) were examined under a scanning electron microscope. The PEx-4 nanospheres were uniform in size (68 ± 3.2 nm) and morphology. (**B**) Plasma Ex-4 concentration between Ex-4 and PEx-4. After subcutaneous administration of Ex-4 or PEx-4, both groups of rats (*n* = 5 each) showed the highest plasma concentrations at 2 h, but PLGA vehicle had no effect on Ex-4 concentration. The peak plasma Ex-4 level 2 h following Ex-4 administration was significantly higher (*p* < 0.05) than that of PEx-4 treatment. (**C**) From 6 h to 7 days, the plasma Ex-4 level of the PEx-4 group was well maintained and significantly (*p* < 0.05) higher than that of the Ex-4 group. (**D**) The maximal CSF Ex-4 level at 2 h after Ex-4 treatment was significantly higher (*p* < 0.05) than that after PEx-4 treatment. However, the CSF Ex-4 concentration of the PEx-4 group was significantly higher (*p* < 0.05) than that of the Ex-4 group 6 to 24 h after subcutaneous administration. (**E**) Hypoglycemic effects of subcutaneously administered Ex-4, PEx-4, or PLGA vehicle in control and diabetes mellitus (DM) rats (*n* = 5 each). In normoglycemic control rats, subcutaneous Ex-4 or PEx-4 significantly reduced blood glucose levels after 6 h of treatment, respectively, but PLGA vehicle nanoparticles did not. PEx-4 significantly (*p* < 0.05) decreased blood glucose levels from 12 h to 7 days in DM rats, but Ex-4 had no effect on DM-evoked hyperglycemia during this period. * *p* < 0.05 between groups using one-way ANOVA, followed by Bonferroni’s post hoc test.

**Figure 2 jpm-12-00390-f002:** Effect of Ex-4 or PEx-4 treatment on CSF RH_2_O_2_ and RHOCl response in the sham or IR rats of control (CON) and diabetes (DM) rats. The typical emission spectra of the chemiluminescence of a test mixture containing H_2_O_2_ and luminol for reference H_2_O_2_ (RH_2_O_2_), HOCl, and luminol for reference HOCl (RHOCl) in Sham, IR, Ex-4-treated IR (IR-Ex4) or PEx-4-treated IR rats (IR-PEx4) of CON and DM rats are displayed in (**A**,**B**). IR increased RH_2_O_2_ (**C**) and RHOCl activity (**D**), whereas DM further enhanced these responses. PEx-4 is more efficient in reducing RH_2_O_2_ and RHOCl activity in the IR-PEx4 group as compared to Ex-4 in the IR-Ex4 group. The statistical difference is indicated by the horizontal line with *p* < 0.05 between groups using one-way ANOVA, followed by Bonferroni’s post hoc test. Each symbol represents one sample in different groups.

**Figure 3 jpm-12-00390-f003:** The panel shows three caudo-rostral projections (CT1, CT2 and CT3) of coronal sections through a rat brain (**A**). The bright zones on T2-weighted images represent 24 h after IR-induced brain edema and are analyzed with ImageJ software (**B**). The percentage (%) of the brain edema zones increased after 24 h of IR and significantly decreased with the pretreatment of Ex-4 (IR-Ex4) or PEx-4 (IR-PEx4) in CT1 (**C**), CT2 (**D**) and CT3 (**E**) sections in both control and DM rat brain coronal sections. The paralleled histologic evidence of CT1 section from eight groups of brain is displayed (**F**). Data are expressed as mean ± SEM in each group (*n* = 6) using the single values. The statistical difference is indicated by the horizontal line with *p <* 0.05 between groups using one-way ANOVA, followed by Bonferroni’s post hoc test. Each symbol represents one sample in different groups.

**Figure 4 jpm-12-00390-f004:** Effect of Ex-4 and PEx-4 on the level of morphological change of prefrontal cortex (**A**) with shrunken morphology indicated by yellow arrows with hematoxylin–eosin stain and CHOP-positive stain ((**C**), green fluorescence indicated by yellow arrows) in rat brain sections of control and DM rats. The number of shrunken neurons is significantly increased after IR injury and decreased in the treatment of Ex-4 (IR-Ex4) or PEx-4 (IR-PEx4) in brain sections (**B**). The number of CHOP-positive cells after IR is significantly increased and decreased in treatment of Ex-4 (IR-Ex4) or PEx-4 (IR-PEx4) in brain sections (**D**). (scale bar = 50 μm). Data are expressed as mean ± SEM in each group (*n* = 6) using the single values. The statistical difference is indicated by the horizontal line with *p <* 0.05 between groups using one-way ANOVA, followed by Bonferroni’s post hoc test. Each symbol represents one sample in different groups.

**Figure 5 jpm-12-00390-f005:** The effect of Ex-4 and PEx-4 treatment on IR-evoked ER stress-, autophagy-, pyroptosis- and apoptosis-associated proteins in Con or DM rat brain (**A**). The levels of ER stress-associated proteins including pIRE-1 (**B**), cleaved Caspase-12 (**C**), pJNK (**D**), CHOP (**E**), autophagy-associated proteins including Beclin-1 (**F**), LC3B (**G**), pyroptosis-associated protein Caspase-1 (**H**), IL-1β (**I**) and apoptosis-associated proteins including cleaved Caspase-3 (**J**) and PARP (**K**) are compared in DM as compared to Con. IR in Con and DM groups further and significantly enhanced pIRE-1, Caspase 12, pJNK, CHOP, Caspase-1, IL-1β, Beclin-1, LC3B and cleaved Caspase-3 and PARP expression as compared to respective Con and DM groups. Ex-4 and PEx-4 treatment with IR injury significantly down-regulated ER stress, pyroptosis, autophagy and apoptosis in the Ex-4-treated IR group (IR-Ex4) and PEx-4-treated IR group (IR-PEx4) of Control and DM rats. Data are expressed as mean ± SEM in each group (*n* = 6) using the single values. The statistical difference is indicated by the horizontal line with *p <* 0.05 between groups using one-way ANOVA, followed by Bonferroni’s post hoc test. Each symbol represents one sample in different groups.

**Figure 6 jpm-12-00390-f006:** Effect of Ex-4 and PEx-4 treatment on the brain expression of pyroptosis, autophagy and apoptosis in control and DM rats subjected to global brain ischemia injury (IR). (**A**): the level of green fluorescent intensity of Caspase-1-mediated pyroptosis, LC3B-mediated autophagy and Caspase-3-mediated apoptosis is determined with an immunofluorescent stain in the rat brains. DM significantly enhanced brain Caspase-1 (**B**), LC3B (**C**) and Caspase-3 (**D**) stains indicated by green fluorescence, whereas IR further and significantly increased Caspase-1, LC3B and Caspase-3 stains in the Con and DM brains as compared to respective Con and DM brains. Ex-4 (IR-Ex4) and PEx-4 (IR-PEx4) treatment significantly reduced IR-enhanced Caspase-1, LC3B and Caspase-3 fluorescence in the brains of control and DM rats. Data are expressed as mean ± SEM in each group (*n* = 6) using the single values. The statistical difference is indicated by the horizontal line with *p <* 0.05 between groups using one-way ANOVA, followed by Bonferroni’s post hoc test. Scale bar = 200 μm. Each symbol represents one sample in different groups.

**Figure 7 jpm-12-00390-f007:** Effect of Ex-4 and PEx-4 on the voiding function in response to stroke injury (IR) among the eight groups of rats. Representative traces of continuous cystometrograms in urethane-anesthetized rats of Control (**A**) or DM groups (**B**). The normal voiding pattern and parameter are demonstrated in Control (Con), whereas a hyposensitive underactive bladder with an increased ICI is indicated in DM bladders. IR caused urine incontinence, depressed ICI, Am, increased MD and BP in Con and DM bladders. The treatment of Ex-4 (IR-Ex4) or PEx-4 (IR-PEx4) seems to play a role in improving IR or DMIR-induced voiding dysfunction. Data are expressed as mean ± SEM in each group (*n* = 6) using the single values. (**C**): ICI, intercontraction interval; (**D**): Am, amplitude; (**E**): MD, micturition duration; (**F**): BP, baseline bladder pressure. The statistical difference is indicated by the horizontal line with *p <* 0.05 between groups using one-way ANOVA, followed by Bonferroni’s post hoc test. Each symbol represents one sample in different groups.

**Figure 8 jpm-12-00390-f008:** The lamina propria layers by H&E stain (indicated by yellow arrows) in DM bladders were thicker than those in control bladders (**A**). Stroke injury (IR) had no effect on the thicken lamina propria layers. Blue collagen fibers (indicated by yellow arrows) stained by Masson’s stain were increased in DM bladder compared with Con bladder (**B**). The use of Image J is applied to analyze the percentage (%) of blue collagen accumulation (**C**). The % of blue collagen stain in each section was significantly higher in the DM bladders than that in Con bladders. IR and the treatment of Ex-4 (IR-Ex4) or PEx-4 (IR-PEx4) had no significant effect on DM-enhanced blue collagen accumulation (**D**). Data are expressed as mean ± SEM in each group (*n* = 6) using the single values. The statistical difference is indicated by the horizontal line with *p <* 0.05 between groups using one-way ANOVA, followed by Bonferroni’s post hoc test. Scale bar (white line) = 200 μm. Each symbol represents one sample in different groups.

**Figure 9 jpm-12-00390-f009:** The effect of Ex-4 and PEx-4 treatment on ER stress-, apoptosis-, pyroptosis- and autophagy-associated proteins in Con or DM rat bladders with or without IR injury (**A**). The levels of ER stress-associated proteins including pIRE-1/IRE-1 ratio (**B**), cleaved Caspase-12 (**C**), pJNK/JNK ratio (**D**), CHOP (**E**), pyroptosis-related proteins including, Caspase-1 (**F**), IL-1β (**G**), autophagy-associated protein Beclin-1 (**H**), LC3B (**I**), and apoptosis-associated proteins including cleaved Caspase-3 (**J**) and PARP (**K**) are up-regulated in DM as compared to Con. IR enhanced pIRE-1/IRE-1, Caspase 12, CHOP, Caspase-1, IL-1β, Beclin-1, LC3B, Caspase 3 and PARP in ConIR as compared to Con. Ex-4 or PEx-4 treatment only efficiently down-regulated CHOP expression in the Con and DM rats of IR-Ex4 and IR-PEx4 groups. Data are expressed as mean ± SEM in each group (*n* = 6) using the single values. The statistical difference is indicated by the horizontal line with *p <* 0.05 between groups using one-way ANOVA, followed by Bonferroni’s post hoc test.

**Figure 10 jpm-12-00390-f010:** Effect of Ex-4 and PEx-4 treatment on the bladder expression of pyroptosis, autophagy and apoptosis in control and DM rats subjected to IR injury. (**A**) the level of green fluorescent intensity of Caspase-1-mediated pyroptosis, LC3B-mediated autophagy and Caspase-3-mediated apoptosis is determined with an immunofluorescent stain in the rat bladders. DM significantly enhanced bladder Caspase-1 (**B**), LC3B (**C**) and Caspase-3 (**D**) stains indicated by green fluorescence, whereas IR further and significantly increased Caspase-1, LC3B and Caspase-3 stains in the IR bladders of CON and DM rats as compared to respective CON and DM bladders. Ex-4 (IR-Ex4) and PEx-4 (IR-PEx4) treatment significantly reduced IR-enhanced Caspase-1, LC3B and Caspase-3 fluorescence in the bladders of CON and DM rats. Data are expressed as mean ± SEM in each group (*n* = 6) using the single values. The statistical difference is indicated by the horizontal line with *p <* 0.05 between groups using one-way ANOVA, followed by Bonferroni’s post hoc test. Scale bar = 200 μm. Each symbol represents one sample in different groups.

## Data Availability

The study did not report any data.

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
