# Peer review of "Long-Lasting Exendin-4-Loaded PLGA Nanoparticles Ameliorate Cerebral Ischemia/Reperfusion Damage in Diabetic Rats"

_jpm, 2022, doi:10.3390/jpm12030390_

Round 1

Reviewer 1 Report

The authors conducted experiments to determine the effects of PEx-4 in comparison with Ex-4 on cerebral ischemia/reperfusion injury associated with micturition center damage-induced cystopathy in diabetic rats. The manuscript was well written in general, while some improvements are still required.

  1. The authors need to discuss the difference of PEx-4 between the studied and those used in the previously published studies (likely by the same research group), such as Chien et al. (2015) J Cereb Blood Flow Metab 35:1790.
  2. There are multiple factors involved in the study, the authors need to define clearly in the statistical analysis the factors and levels of each factors, and thus the model for the analysis.
  3. Was appropriate to use t-test within each individual group? Please define clearly what factor was for each group.
  4. How were the parameters with multiple measurements analysed?
  5. Some improvements in writing, especially grammar and sentence structure are required before acceptance. I have highlighted some sentences, words, and phrases as suggestions. See the attached file.
  6. Conclusion could be expanded to cover the full aspects that were supported by the results. 

Author Response

Response to the Reviewer 1:

Comments and Suggestions for Authors

The authors conducted experiments to determine the effects of PEx-4 in comparison with Ex-4 on cerebral ischemia/reperfusion injury associated with micturition center damage-induced cystopathy in diabetic rats. The manuscript was well written in general, while some improvements are still required.

  1. The authors need to discuss the difference of PEx-4 between the studied and those used in the previously published studies (likely by the same research group), such as Chien et al. (2015) J Cereb Blood Flow Metab 35:1790.

Ans: There are several differences between the present study and the previous report of Chien et al. (2015).

  1. The present study using Ex-4 nanoparticle (PEx-4, 68.0 ± 3.2 nm) was different from the PEx-4 microspheres (15 ± 3.3 µm) in Chien et al., 2015 for a different size. Because PEx-4 microspheres (15 ± 3.3 µm) may have an effect to interrupt capillary blood flow.
  2. The present study evaluated the PEx-4 effect on three types of programmed cell death including autophagy, apoptosis and pyroptosis in ischemic brain and post-IR bladder function, which was different from the previous study in evaluating ischemic brain with Akt/eNOS mediated microcirculation and neurologic deficits.
  3. The present study evaluated the PEx-4 effect on CSF ROS in ischemic brain, which was different from the previous study in evaluating ROS from ischemic brain surface.

  1. There are multiple factors involved in the study, the authors need to define clearly in the statistical analysis the factors and levels of each factors, and thus the model for the analysis.

Ans: The manuscript evaluated the determined parameter among control, diabetes, IR, Ex-4 and PEx-4 groups. Statistical analyses were performed using one-way ANOVA, followed by Bonferroni's post-hoc test. Differences between groups were considered statistically significant when P < 0.05.

  1. Was appropriate to use t-test within each individual group? Please define clearly what factor was for each group.

Ans: Thanks for your comments. We have changed the statistical methods and described in the Figure legend. The use of t-test within each individual group was not appropriate. Therefore, we have altered the statistical methods in the revised manuscript.

Statistical analyses were performed using one-way ANOVA, followed by Bonferroni's post-hoc test. Differences between groups were considered statistically significant when P < 0.05.

  1. How were the parameters with multiple measurements analysed?

Ans: Thanks for your comments. We have altered the statistical methods in the revised manuscript and described in each Figure legend. The manuscript evaluated the determined parameter among control, diabetes, IR, Ex-4 and PEx-4 groups. Statistical analyses were performed using one-way ANOVA, followed by Bonferroni's post-hoc test. Differences between groups were considered statistically significant when P < 0.05.

In the Methods section: “All values and data were expressed as mean ± standard error mean (SEM). Data recording was not blinded because of technical requirements, but data analyses were blinded for biochemical analyses. Statistical analyses were performed using one-way ANOVA, followed by Bonferroni's post-hoc test. Differences between groups were considered statistically significant when P < 0.05.”

  1. Some improvements in writing, especially grammar and sentence structure are required before acceptance. I have highlighted some sentences, words, and phrases as suggestions. See the attached file.

Ans: Thanks for your comments. We have corrected the mistakes and improved the writing format, grammar and sentence structure by an expert in English.

  1. Conclusion could be expanded to cover the full aspects that were supported by the results.

Ans: We have expanded the conclusion as you mentioned in the revised manuscript.

In the Conclusion section: According to our data, T1DM and IR significantly enhanced CSF RH2O2, and pIRE-1/caspase-12/pJNK/CHOP-mediated ER stress, caspase-3/PARP mediated apoptosis, Beclin-1/LC3B-mediated autophagy and caspase-1/IL-1β-mediated pyroptosis signaling in the damaged brains. PEx-4 were more efficient than Ex-4 in attenuating IR-evoked prefrontal cortex edema, the impairment in micturition center and the enhanced level of CSF RH2O2 and HOCl, ER stress, apoptosis, autophagy and pyroptosis parameters in the damaged brains, but less effect on IR-induced voiding dysfunction in bladders of T1DM rats. In conclusion, these results suggest that PEx-4 with stronger antioxidant activity and long-lasting bioavailability may confer efficiently therapeutic efficacy to ameliorate IR-evoked brain damage through the inhibitory action on oxidative stress, ER stress, apoptosis, autophagy and pyroptosis signaling in diabetic rats. We implicate that long-lasting release of GLP-1 agonists PEx-4 may confer therapeutic potential to treat cerebral IR injury in the DM subjects in clinical trials.

Reviewer 2 Report

The rationale for the study is unclear as the introduction is confusing, comprising several pieces of apparently unlinked information. A more integrated appraisal of the relevant literature would be appropriate to provide the context for the study.

Immunoblots: show the MW markers for actin.

A dimensional bar is missing in several figures.

Figure 8: "Stain intensity" is not a quantitative measure; please amend.

The manuscript is mainly descriptive and focused on its (not fully supported) conclusions, not adequately acknowledging the limitations of the study. The strengths and limitations of the study should be deeply addressed, taking into account sources of potential bias or imprecision: Discuss both direction and magnitude of any potential bias.

The following pertinent reports should be mentioned:

PMID: 35046890

PMID: 34948054 

PMID: 34943088

PMID: 34830136 

PMID: 34658880

PMID: 34539414

It is advisable to the Authors to incorporate a pictorial or cartoon representation of the main results of the study to increase the overall impact of the manuscript.

Author Response

Response to the Reviewer 2:

Comments and Suggestions for Authors

The rationale for the study is unclear as the introduction is confusing, comprising several pieces of apparently unlinked information. A more integrated appraisal of the relevant literature would be appropriate to provide the context for the study.

  1. Immunoblots: show the MW markers for actin.

Ans: We have included the 42 kD molecular weight of b-actin in the revised Figure 5 and Figure 9.

  1. A dimensional bar is missing in several figures.

Ans: We have included the dimension bar in all figures.

  1. Figure 8: "Stain intensity" is not a quantitative measure; please amend.

Ans: We have corrected with “% of blue collagen stain” in the revised Figure 8.

  1. The manuscript is mainly descriptive and focused on its (not fully supported) conclusions, not adequately acknowledging the limitations of the study. The strengths and limitations of the study should be deeply addressed, taking into account sources of potential bias or imprecision: Discuss both direction and magnitude of any potential bias.

Ans: Thanks for your comments. We have modified the manuscript and increased several papers to address the strengths and mechanisms in the revised manuscript. These can be seen in the Discussion section.

In the Discussion section: the first paragraph

“The increasing use of nanomaterials in a variety of industrial, commercial, medical products, and their environmental spreading has raised concerns regarding their potential toxicity on human health. For example, titanium dioxide nanoparticles may induce ultrastructural alterations appeared at the third week by the inducing activation of caspase-3 and increase intracellular ROS and DNA damage [24]. On the other hand, PLGA nanoparticles were beneficial for ensuring structural stability, enhancing absorption, followed by sustained drug release [25]. Biodegradable nanoparticles like carbohydrate polymeric PLGA are frequently used to improve the therapeutic value of various water soluble/insoluble medicinal drugs and bioactive molecules by improving bioavailability, solubility and retention time [15]. A specific drug loaded with PLGA nanoparticles using the single emulsion-solvent evaporation method possessed high drug loading content, encapsulation efficiency, a small size, and negative surface charge contributed to a specific intracellular accumulation [26]. Previous results also suggest that PLGA nanoparticles represent a high potential for utilization in in anticancer therapy due to their effectiveness and safety [27]. PLGA nanoparticles loaded with miR-30b-5p can improve cardiac function, attenuate myocardial injury, and regulate the expression of factors associated with cardiac hypertrophy and inflammation by targeting TGFBR2 [28]. PLGA nanoparticles are well known as promising delivery systems, especially in the area of nose-to-brain delivery by transcellular, intracellular and paracellular transport [29]. The intracellular uptake was observed for 80 and 175 nm within only 5 min after application to the epithelium. These information implicated that PLGA nanoparticles with high efficacy, small size, no toxicity and easy uptake by cells are excellent delivery systems.”

In the second paragraph of Discussion section:

“In addition, our unpublished data indicated the dose of PEx-4 or Ex-4 did not have nephrotoxicity to impair renal tissue or renal function because the serum creatinine was not increased. Since Ex-4 was administered weekly despite of the very short half life, the maximum efficacy of this drug might be not fully achieved in this study. A dosage of 50 μg/kg/week used in this study could be low and could not obtain adequate protective efficacy in this study. Thus, the superior efficacy of PEx-4 than Ex-4 with different dosages or treatment frequency requires further studies to conclude their therapeutic potential.”

In the third paragraph of Discussion section:

“It has revealed that lesions of the medial prefrontal cortex caused nocturnal incontinence, urinary urgency and frequent micturition and only rarely caused urinary retention [34,35]. In our study, the ischemic brain with the edema of prefrontal cortex disturbed micturition and maintenance of urinary continence in rat. Besides, lesioning the medial prefrontal cortex would prolong the time interval between volume-evoked bladder contractions without changing the amplitude of contractions [36]. Our brain IR injury induced bladder dysfunction data was consistent with these previous studies. We evaluated the ROS amount in the IR brain and found that a significant increase of ROS amount in the CSF indicating the toxic effect of ROS in the damaged brain. The two major ROS generated from activated neutrophils via the myeloperoxidase system are hydrogen peroxide (H2O2) and hypochlorite (HOCl) [19], which are important mediators in oxidative stress and inflammation. Our data found that DM and IR injury increased H2O2 and HOCl in the CSF implicating neuronal inflammation and oxidative stress. The novel finding indicated that increased ROS in CSF may impair brain function and micturition center leading to voiding dysfunction. A cross-talk role of exacerbated ROS production may be occurred between brain and bladder. Increased excess oxidative stress evoked neural and vascular injury contributed to DM evoked stroke associated with voiding dysfunction [2-5]. Exacerbated ROS production leads to tissue damage through ROS-evoked abnormal signal transduction, inflammatory monocyte/macrophage infiltration, cellular dysfunction and programmed cell death (autophagy, apoptosis and pyroptosis) in several kinds of cells. ROS generation from the NADPH (nicotinamide adenine dinucleotide phosphate) oxidase or mitochondria may induce apoptosis, autophagy, or pyroptosis, via execution by caspases, lysosomal proteases, or endonucleases [37]. The increased ROS trigger apoptosis by activating Bax expression/caspase 3 activity/poly-(ADP-ribose)-polymerase (PARP) fragments, enhance autophagy by activating Beclin-1/Atg5-Atg12/LC3-II pathway and/or evoke pyroptosis via the activation of caspase 1/IL-1β signaling [37]. In our western blot and immunofluorescent data, caspase-1 mediated pyroptosis, LC3B mediated autophagy and caspase-3 mediated apoptosis were significantly enhanced in the IR brain and DM bladder. Our prepared PEx-4 was more efficient than Ex-4 in reducing these oxidative stress (CSF ROS), ER stress and three types of programmed cell death in IR and DM induced injury implicating the therapeutic potential of PEx-4 in the reduction of oxidative stress, ER stress, apoptosis, autophagy and pyroptosis.”

  1. The following pertinent reports should be mentioned:

Ans: The following papers have been cited and discussed in the revised manuscript.

  1. Mancuso, F.; Arato, I.; Di Michele, A.; Antognelli, C.; et al. Effects of titanium dioxide nanoparticles on porcine prepubertal sertoli cells: An "In Vitro" study. Front Endocrinol (Lausanne) 2022, 12, 751915.
  2. Akel, H.; Csóka, I.; Ambrus, R.; Bocsik, A.; et al. In vitro comparative study of solid lipid and PLGA nanoparticles designed to facilitate nose-to-brain delivery of insulin. Int J Mol Sci 2021, 22(24), 13258.
  3. Mollaeva, M.R.; Nikolskaya, E.; Beganovskaya, V.; Sokol, M.; et al. Oxidative damage induced by phototoxic pheophorbide a 17-diethylene glycol ester encapsulated in PLGA nanoparticles. Antioxidants (Basel) 2021;10(12):1985.
  4. Mollaeva, M.R.; Yabbarov, N.; Sokol, M.; Chirkina, M.; et al. Optimization, characterization and pharmacokinetic study of meso-tetraphenylporphyrin metal complex-loaded PLGA nanoparticles. Int J Mol Sci 2021, 22(22), 12261.
  5. Ren, Y.; Wang, X.; Liang, H.; He, W.; et al. Mechanism of miR-30b-5p-loaded PEG-PLGA nanoparticles for targeted treatment of heart failure. Front Pharmacol 2021, 12, 745429.
  6. Spindler, L.M.; Feuerhake, A.; Ladel, S.; Günday, C.; et al. Nano-in-micro-particles consisting of PLGA nanoparticles embedded in chitosan microparticles via spray-drying enhances their uptake in the olfactory mucosa. Front Pharmacol 2021,12, 732954.

In the Discussion section:

“The increasing use of nanomaterials in a variety of industrial, commercial, medical products, and their environmental spreading has raised concerns regarding their potential toxicity on human health. For example, titanium dioxide nanoparticles may induce ultrastructural alterations appeared at the third week by the inducing activation of caspase-3 and increase intracellular ROS and DNA damage [24]. On the other hand, PLGA nanoparticles were beneficial for ensuring structural stability, enhancing absorption, followed by sustained drug release [25]. Biodegradable nanoparticles like carbohydrate polymeric PLGA are frequently used to improve the therapeutic value of various water soluble/insoluble medicinal drugs and bioactive molecules by improving bioavailability, solubility and retention time [15]. A specific drug loaded with PLGA nanoparticles using the single emulsion-solvent evaporation method possessed high drug loading content, encapsulation efficiency, a small size, and negative surface charge contributed to a specific intracellular accumulation [26]. Previous results also suggest that PLGA nanoparticles represent a high potential for utilization in in anticancer therapy due to their effectiveness and safety [27]. PLGA nanoparticles loaded with miR-30b-5p can improve cardiac function, attenuate myocardial injury, and regulate the expression of factors associated with cardiac hypertrophy and inflammation by targeting TGFBR2 [28]. PLGA nanoparticles are well known as promising delivery systems, especially in the area of nose-to-brain delivery by transcellular, intracellular and paracellular transport [29]. The intracellular uptake was observed for 80 and 175 nm within only 5 min after application to the epithelium. These information implicated that PLGA nanoparticles with high efficacy, small size, no toxicity and easy uptake by cells are excellent delivery systems.“

  1. It is advisable to the Authors to incorporate a pictorial or cartoon representation of the main results of the study to increase the overall impact of the manuscript.

Ans: Thanks for your comments. We have incorporated a graphic abstract to represent the main results of this study to increase the overall impact as you mentioned in the revised manuscript.

Round 2

Reviewer 2 Report

-